# Few-Cost Salient Object Detection with Adversarial-Paced Learning

**Dingwen Zhang[1], Haibin Tian[1] and Jungong Han[2✉]**
[1]School of Mechano-Electronic Engineering, Xidian University, Xi'an, Shaanxi 710071
[2]Computer Science Department, Aberystwyth University, Ceredigion, SY23 3FL
`zdw@xidian.edu.cn, haibintian@foxmail.com, jungonghan77@gmail.com`

## Abstract

Detecting and segmenting salient objects from given image scenes has received great attention in recent years. A fundamental challenge in training the existing deep saliency detection models is the requirement of large amounts of annotated data. While gathering large quantities of training data becomes cheap and easy, annotating the data is an expensive process in terms of time, labor and human expertise. To address this problem, this paper proposes to learn the effective salient object detection model based on the manual annotation on a few training images only, thus dramatically alleviating human labor in training models. To this end, we name this task as the few-cost salient object detection and propose an adversarial-paced learning (APL)-based framework to facilitate the few-cost learning scenario. Essentially, APL is derived from the self-paced learning (SPL) regime but it infers the robust learning pace through the data-driven adversarial learning mechanism rather than the heuristic design of the learning regularizer. Comprehensive experiments on four widely-used benchmark datasets demonstrate that the proposed method can effectively approach to the existing supervised deep salient object detection models with only 1k human-annotated training images. The project page is available at `https://github.com/hb-stone/FC-SOD`.

## 1 Introduction

With the goal of automatically discovering object regions that attract human attention from the given image scenes, salient object detection has become prevalent in the computer vision community [1–3]. Due to its wide range of applications, large amounts of efforts have been made to build powerful deep convolutional network models for addressing this problem. Relying on the large-scale human-annotated training image data, methods presented in recent years have been undergoing unprecedentedly rapid development. However, as it is often time-consuming and expensive to provide the manually annotated pixel-wise ground-truth annotation, the training processes of the most existing methods are costly in terms of time and money. To this end, this paper studies the challenge to learn an effective salient object detection model by only using the manual annotation on a few training images.

Inspired by the few-shot learning problems [4, 5] that use only a few training samples of the targets, we name the investigated problem as the few-cost salient object detection (FC-SOD) problem as it costs only a few annotated training samples. More specifically, in FC-SOD, the scenario is that the training data contain the large scale training images, but only a few of them have the pixel-wise annotation on the salient objects. Such a problem sounds also similar to a semi-supervised learning problem. As [6] has defined the semi-supervised SOD (SS-SOD) task as the task to partially label the regions within each image firstly and then use both the labeled and unlabeled regions to learn the saliency model, we define the task considered in this work as FC-SOD to avoid confusion.

When designing the few-cost learning framework for salient object detection, the key problem is to progressively annotate the unlabeled training images according to the knowledge mined from the small scale annotated training images. However, such a learning procedure may turn to trivial solutions when noisy or wrong annotations are set to the unlabeled training images and introduced into the intermediate learning process. A simple yet effective way to alleviate this problem is the use of the self-paced learning (SPL) mechanism [7] in the few-cost learning framework as SPL is inherently a robust learning mechanism that helps the learning system explore the samples containing truthful knowledge (i.e., those with accurate labels) while screening the samples with unreliable knowledge (i.e., those with noisy or wrong labels). Such robust learning capacity has also been proved by recent studies in vision tasks [8–12]. For example, [13, 14] apply the SPL process to refine the saliency maps obtained in co-saliency detection. [15, 16] integrate SPL and adversarial learning for domain adaption and clustering, respectively.

At the core of SPL is the design of the self-paced regularizer, based on which the learner can dynamically assign proper learning weights to the samples—reliable labels are assigned with large learning weights while noisy labels are attached with small learning weights—and this dynamical weighting process leads to the robust learning pace to guide the learning procedure. Currently, the main strategy of SPL methods designs the self-paced regularizer based on human knowledge in the corresponding task domain. This strategy is, to some extent, heuristic and may lead to suboptimal solutions due to the insufficient exploration of the data. Thus, a more reasonable way to infer the robust learning pace might be in a data-driven manner, where the concrete formulation of the learning weights is learned from the data rather than being manually designed by humans. In this way, the learning system can, on one hand, alleviate its dependency on manual design, which endows stronger learning capacity to the learner. On the other hand, by leveraging the samples from the corresponding task domain, it can obtain the more suitable learning pace for any task under investigation.

To implement such a learning mechanism, we propose a novel adversarial-paced learning framework, which is derived from the conventional SPL and driven by the underlying relationship between the optimization processes of SPL and the well-known generative adversarial learning (GAL) [17]. Specifically, it is known that the alternative optimization strategy commonly used to solve the SPL problem can be considered as the majorization minimization algorithm that is implemented on a latent SPL objective function [18]. While the optimization process of GAL is also a min-max game. Thus, the proposed APL framework can be implemented with a similar optimization process as the conventional SPL methods but with a different data-driven mechanism to infer the learning weights and generate the learning pace. In APL, the adversarial learning mechanism will enable the learner to tell which of the predicted labels are "real", i.e., reliable, while which are not.

To sum up, this work mainly contains the following three-fold contributions:

- We explore an under-studied task called few-cost salient object detection. Compared with the conventional fully supervised salient object detection, it requires only the manual annotation on a small number of training images and thus can alleviate the annotation cost for training deep salient object detectors.

- We reveal the underlying relationship between SPL and GAL to establish a novel adversarial-paced learning framework. By implicitly encoding the pace regularizer in an additional model called pace-generator, APL can infer the robust learning pace through a data-driven adversarial learning mechanism rather than the heuristic design of the learning regularizer.

- Comprehensive experiments on widely-used benchmark datasets have been implemented to evaluate the effectiveness of the proposed approach. Notably, by using the annotation of only 1k training images, the proposed approach outperforms the existing un-/semi-/weakly supervised SOD approaches and performs comparably to the fully supervised SOD models.

## 2 Previous Works

**Salient Object Detection.** In light of the advanced development in deep learning, recent salient object detection methods mainly adopted the CNN models to learn saliency patterns under a fully supervised fashion. Most of these methods, e.g., [19–24], focus on extracting representative deep features in more effective and efficient ways. For learning strong feature representations, a new trend in this field is appeared, which provides richer supervision to guide the network learning process. Under this strategy, some recent works introduce the human-annotated contour information into

the network learning process. For example, Wu et al. [25] integrated salient object detection and foreground contour detection tasks in an intertwined manner, which enables the learned model to generate saliency maps with uniform highlight regions. Liu et al. [26] built simple yet effective pooling-based modules to decode the deep features to infer both saliency maps and contour maps.

Different from the above-mentioned direction, this work explores an alternative direction to advance the SOD community—Instead of acquiring richer supervision, this work studies how to shrink the supervision. Research from this direction could dramatically reduce the labor costs and endow the deep model stronger learning capacity. Notice that there is also a small number of works [27–29] that share the similar spirit with this work. However, the problems considered by them are under the weakly supervised or unsupervised learning scenarios, which are distinct from our investigated few-cost learning problem. Unlike our work, Yan et al. [30] define their problem on sparsely annotated video frames. They train video salient object detector by leveraging the dependencies among adjacent video frames. Notice that the upper bound of FCSOD should be higher than unsupervised SOD as unsupervised SOD does not use any human annotation. With the same label cost, the upper bound of FCSOD and weakly supervised SOD (WSSOD) should be close. However, as FCSOD leverages a small number of strong annotations while WSSOD leverages a lot of weak annotations, FCSOD should theoretically work better when dealing with data with small domain shifts.

**Semi-supervised GAL.** The proposed APL is also related to the semi-supervised GAL (SS-GAL) framework, such as [31–33], as both of them use the labeled data and unlabeled data in the learning procedure. However, the generators and discriminators used in APL and SS-GAL play very distinct functions—In SS-GAL, the generator is used to generate extra training samples from the input noise signals while the discriminator acts as a multi-class classifier to predict labels for the input training samples. In contrast, the generator in APL is used to predict labels for the input training samples while the discriminator is used to judge whether the input label is with a realistic structure. The core difference between APL and SS-GAL is that the unlabeled data in APL are used to learn the inference function for label weighting whereas the unlabeled data in SS-GAL are used to learn the mapping function for feature representation. The work of [34] is also related to this work, where an IoT-oriented saliency learning framework is presented with the intention to leverage both labeled and unlabeled data from different problem domains for training. In contrast, our work aims to minimize the annotation cost for saliency learning in a single domain.

Another interesting SS-GAL framework is proposed by [35], which is intuitively analogous with our approach. However, from the perspective of the high-level idea, our work differs from [35] by deriving APL from the SPL regime and establishing the learning framework in a theoretically-sound manner rather than a heuristic manner. While from the perspective of implementation details, our works differ from [35] by proposing a novel global structure-guided pixel weighting scheme and designing the different objective function and optimization strategy. By revealing the underlying relationship between SPL and GAL mechanism and explicitly modeling the reliability weights $\mathbf{V}$ (see Eq.2), this work could provide a theoretical explanation and a new interpretation of the learning framework of [35].

It is also worth mentioning that when comparing the semi-supervised semantic segmentation (SSS) problem and the investigated few-cost salient object detection problem, besides the superficial difference in the number of classes, the challenges met by them are also different. Specifically, as the salient class that needs to be segmented in FCSOD would usually cover a number of different semantics rather than forming by a specific semantic, FCSOD is encountered with heavier intra-class variance than SSS. This would bringing the challenging learning ambiguity issue in FCSOD. Besides, the current SSS methods usually leverage the semantic vector as an informative attribute to guide the GAN-based semi-supervised learning process, which, however, is absent from the FCSOD task. Consequently, such SSS algorithms could not be easily applied to FCSOD.

## 3    Adversarial-Paced Learning

**Formulation.** Given a small-scale collection $\mathcal{D}^l$, which consists of the manually labeled training images $\{\mathbf{X}_i^l, \mathbf{Y}_i^l\}$, and a larger-scale collection $\mathcal{D}^u$ that contains the unlabeled training images $\{\mathbf{X}_j^u\}$, we denote the pseudo label of $\mathbf{X}_j^u$ denotes as $\mathbf{Y}_j^u$, which needs to be inferred during the learning process. To solve this problem, the existing methods may adopt the SPL [7]-based learning framework to infer the pseudo labels for the unlabeled images and then involve the self-paced regularizer to guide a robust learning procedure to against the noise brought by the inaccurate pseudo label. Such

**Algorithm 1** The conventional SPL scheme. $g(\cdot)$ and $f(\cdot)$ denote the learning model and the self-paced regularizer, respectively. Notice that we only show the main objective function and ignore the constraints here.

---
**Input:** The set of labeled samples $\{\mathbf{X}_i^l, \mathbf{Y}_i^l\}$; The set of unlabelled samples $\mathbf{X}_i^u$;
**Output:** Model parameter $\Theta$;
1: Given objective function:
2: $\quad \min\limits_{\Theta, \mathcal{V}, \mathcal{Y}^u} \mathcal{L}^l(\Theta) + \mathcal{L}^u(\mathcal{Y}^u, \Theta, \mathcal{V}) + \lambda f(\mathcal{V})$
3: Initializing $\Theta$ and $\mathcal{V}$;
4: **repeat**
5: $\quad$ Update $\mathcal{Y}^u$ with fixed $\Theta$ and $\mathcal{V}$:
6: $\quad\quad$ Calculate $g(\mathbf{X}_j^u|\Theta)$
7: $\quad$ Update $\mathcal{V}$ with fixed $\Theta$ and $\mathcal{Y}^u$:
8: $\quad\quad$ Optimize $\min_{\mathcal{V}} \mathcal{L}^u(\mathcal{Y}^u, \Theta, \mathcal{V}) + \lambda f(\mathcal{V})$
9: $\quad$ Update $\Theta$ with fixed $\mathcal{Y}^u$ and $\mathcal{V}$:
10: $\quad\quad$ Optimize $\min\limits_{\Theta} \mathcal{L}^l(\Theta) + \mathcal{L}^u(\mathcal{Y}^u, \Theta, \mathcal{V})$
11: **until** converge
12: **return** Model parameter $\Theta$;

---

**Algorithm 2** The proposed APL scheme. $T(\cdot)$ and $P(\cdot)$ denote the task-predictor and the pace-generator, respectively. Notice that we only show the main objective function and ignore the constraints here.

---
**Input:** The set of labeled samples $\{\mathbf{X}_i^l, \mathbf{Y}_i^l\}$; The set of unlabelled samples $\mathbf{X}_j^u$;
**Output:** Model parameter $\Theta = \{\Psi, \Phi\}$;
1: Given objective function:
2: $\quad \min\limits_{\Psi, \mathcal{V}, \mathcal{Y}^u} \max\limits_{\Phi} \mathcal{L}^l(\Psi) + \mathcal{L}^u(\mathcal{Y}^u, \Psi, \mathcal{V}) + \beta \mathcal{L}^p(\mathcal{Y}^u, \Psi, \Phi)$
3: Initializing $\Theta$ and $\mathcal{V}$;
4: **repeat**
5: $\quad$ Update $\mathcal{Y}^u$ with fixed $\Theta = \{\Psi, \Phi\}$ and $\mathcal{V}$:
6: $\quad\quad$ Calculate $T(\mathbf{X}_j^u|\Psi)$
7: $\quad$ Update $\mathcal{V}$ with fixed $\Theta = \{\Psi, \Phi\}$ and $\mathcal{Y}^u$:
8: $\quad\quad$ Calculate $P(T(\mathbf{X}_j|\Psi)|\Phi)$
9: $\quad$ Update $\Theta = \{\Psi, \Phi\}$ with fixed $\mathcal{Y}^u$ and $\mathcal{V}$:
10: $\quad\quad$ $\min\limits_{\Psi} \max\limits_{\Phi} \mathcal{L}^l(\Psi) + \mathcal{L}^u(\mathcal{Y}^u, \Psi, \mathcal{V}) + \beta \mathcal{L}^p(\Psi, \Phi)$
11: **until** converge
12: **return** Model parameter $\Theta = \{\Psi, \Phi\}$;

---

learning mechanism can be formulated as:

$$\min\limits_{\Theta, \mathcal{V}, \mathcal{Y}^u} \mathcal{L}^l(\Theta) + \mathcal{L}^u(\mathcal{Y}^u, \Theta, \mathcal{V}) + \lambda f(\mathcal{V}), \tag{1}$$

where $\Theta$ denotes the parameters of the learning model. $\mathcal{V} = \{\mathbf{V}_1, \mathbf{V}_2, \cdots, \mathbf{V}_M\}$ and $\mathcal{Y}^u = \{\mathbf{Y}_1^u, \mathbf{Y}_2^u, \cdots, \mathbf{Y}_M^u\}$ indicate the collections of the inferred reliability weight matrixes and the pseudo labels (binary and structured) for the unlabeled training images. $\mathcal{L}^l$ and $\mathcal{L}^u$ indicate the loss functions for labeled and unlabeled data, respectively. $f(\mathcal{V})$ is the self-paced regularizer which is usually designed based on the human knowledge in the corresponding task domain. For example, Jiang et al. [9] proposed the linear soft weighting regularizer, logarithmic soft weighting regularizer, and mixture regularizer weighting to build the self-paced re-ranking model for multimedia search. The self-paced regularizer proposed by Zhang et al. [36] consists of a $\ell_1$-norm, a $\ell_{0.5,1}$-norm, and a Laplacian term for considering the group property in co-saliency detection. Li et al. [37] introduced a $\ell_1$-norm and an adaptive $\ell_{2,1}$-norm into the self-paced regularizer to simultaneously explore the task complexity and instance complexity for multi-task learning.

Unlike the aforementioned SPL regime, this paper explores a new data-driven strategy, named as APL, to infer the robust learning pace, where the label reliability inference mechanism is implicitly encoded by an additional model, which we call the pace-generator, with undefined but learnable functions. Consequently, APL is equipped with both a task-predictor and a pace-generator, where the task-predictor $T(\cdot)$ generates the task-oriented prediction for the input image and is used to predict the pseudo labels for the unlabeled training images. While the pace-generator $P(\cdot)$ discriminates the reliable and unreliable labels from the obtained pseudo labels dynamically to form a robust learning pace. Then, the whole learning objective function of APL becomes:

$$\begin{cases} \min\limits_{\Psi, \mathcal{V}, \mathcal{Y}^u} \max\limits_{\Phi} \mathcal{L}^l(\Psi) + \mathcal{L}^u(\mathcal{Y}^u, \Psi, \mathcal{V}) + \beta \mathcal{L}^p(\Psi, \Phi), \\ s.t. \ \mathbf{V}_j = P(T(\mathbf{X}_j^u|\Psi)|\Phi), \forall j = 1, 2, \cdots, M, \end{cases} \tag{2}$$

where $\Psi$ and $\Phi$ denote the model parameters of the task-predictor and the pace-generator, respectively. By introducing the pace-generator $P(\cdot)$ to infer the reliability of the generated task-oriented labels $T(\mathbf{X}_j^u|\Psi)$ on the unlabeled training images, we have $\mathbf{V}_j = P(T(\mathbf{X}_j^u|\Psi)|\Phi)$. $\mathcal{L}^p$ indicates the objective function for inferring the learning pace under the adversarial-paced learning mechanism, which replaces the self-paced regularizer in Eq.1. $\beta$ is the free parameters to weigh $\mathcal{L}^p$. With the underlying relationship between the optimization processes of SPL and GAL, the proposed learning framework can be optimized under a similar pipeline to the conventional SPL methods (see Alg. 1 and Alg. 2) but is able to infer meaningful learning paces through the adversarial-learned pace-generator.

Specifically, we adopt the commonly used cross-entropy loss in $\mathcal{L}^l$ to measure the consistency between the predicted saliency masks and the corresponding human-annotated ground-truth masks:

$$\mathcal{L}^l = -\sum\nolimits_{i \in \mathcal{D}^l} \Gamma[(\mathbf{1} - \mathbf{Y}_i^l) \log(\mathbf{1} - T(\mathbf{X}_i^l|\Psi)) + \mathbf{Y}_i^l \log T(\mathbf{X}_i^l|\Psi)], \tag{3}$$

where $\Gamma[\cdot]$ indicates the operation to sum all the elements in the input matrix. Different from $\mathcal{L}^l$, $\mathcal{L}^u$ is defined as a weighted cross-entropy loss, which utilizes the pseudo task-oriented labels generated by the task-predictor as well as the inferred reliability weights as the supervision:

$$\mathcal{L}^u = -\sum\nolimits_{j \in \mathcal{D}^u} \Gamma[\mathbf{V}_j \left((\mathbf{1} - \mathbf{Y}_j^u) \log(\mathbf{1} - T(\mathbf{X}_j^u|\Psi)) + \mathbf{Y}_j^u \log T(\mathbf{X}_j^u|\Psi))\right], \qquad (4)$$

where the element-wise product is used between any two matrixes and it goes the same for all other equations in this paper. To learn a robust learning pace under a data-driven adversarial learning mechanism, we introduce a pace-generator $P(\cdot)$ and define $\mathcal{L}^p$ as:

$$\begin{aligned}
\mathcal{L}^p = \sum\nolimits_{i \in \mathcal{D}^l} \Gamma[\log P(\mathbf{Y}_i^l|\Phi)] + \sum\nolimits_{i \in \mathcal{D}^l} \Gamma[\log(\mathbf{1} - P(T(\mathbf{X}_i^l|\Psi)|\Phi))] \\
+ \eta \sum\nolimits_{j \in \mathcal{D}^u} \Gamma[\log(\mathbf{1} - P(T(\mathbf{X}_j^u|\Psi)|\Phi))].
\end{aligned} \qquad (5)$$

By minimizing $\mathcal{L}^p$, the task-predictor is trained to predict the high-quality task-oriented labels so that the pace-generator would recognize them as the realistic ones, i.e., making $P(T(\mathbf{X}_i^l|\Psi)|\Phi)$ and $P(T(\mathbf{X}_j^u|\Psi)|\Phi)$ close to 1. While by maximizing $\mathcal{L}^p$, the pace-generator can be trained to discriminate between the generated fake task-oriented labels and the real human annotation, i.e., making $P(\mathbf{Y}_i^l|\Phi)$ close to 1 while $P(T(\mathbf{X}_i^l|\Psi)|\Phi)$ and $P(T(\mathbf{X}_j^u|\Psi)|\Phi)$ close to 0, so that it can acquire the capacity to measure the reliability and truthfulness of the predicted labels.

**Optimization.** Firstly, we initialize the model parameters $\{\Phi, \Psi\}$ by training the task-predictor and the pace-generator on the labeled training data under a common generative-adversarial learning mechanism:

$$\min_{\Psi} \max_{\Phi} \mathcal{L}^l(\Psi) + \beta(\sum\nolimits_{i \in \mathcal{D}^l} \Gamma[\log P(\mathbf{Y}_i^l|\Phi)] + \sum\nolimits_{i \in \mathcal{D}^l} \Gamma[\log(1 - P(T(\mathbf{X}_i^l|\Psi)|\Phi))]). \quad (6)$$

Following the standard GAN training procedure, we adopt a two-stage learning approach to optimize Eq. 6: In the first stage, we learn the parameters of the pace-generator by fixing $\Psi$. In this case, $\Phi$ can be optimized by maximizing $\sum_{i \in \mathcal{D}^l} \Gamma[\log P(\mathbf{Y}_i^l|\Phi)] + \sum_{i \in \mathcal{D}^l} \Gamma[\log(1 - P(T(\mathbf{X}_i^l|\Psi)|\Phi))]$. While in the second stage, we learn the parameters of the task-predictor by fixing $\Phi$. In this case, $\Psi$ can be optimized by minimizing $\mathcal{L}^l(\Psi) + \beta \sum_{i \in \mathcal{D}^l} \Gamma[\log(1 - P(T(\mathbf{X}_i^l|\Psi)|\Phi))]$. The reliability weights in $\{\mathbf{V}_j\}$ are initialized as ones.

After the initialization process, we alternatively infer $\mathcal{Y}^u$, $\mathcal{V}$ and learn $\{\Phi, \Psi\}$ in each learning iteration. Specifically, we first infer $\mathcal{Y}^u$ and $\mathcal{V}$ based on the network models learned from the previous learning iteration, where $\mathcal{Y}^u$ is obtained by minimizing $\mathcal{L}^u(\mathcal{Y}^u, \Psi, \mathcal{V})$, i.e., forwarding the training images through the learned task-predictor and then binarizing the obtained outputs via the threshold of 0.5, while $\mathcal{V}$ is obtained by following the definition in Eq. 2, i.e., $\mathbf{V}_j = P(T(\mathbf{X}_j^u|\Psi)|\Phi)$. Then, we learn the network parameters $\{\Phi, \Psi\}$ based on the inferred $\mathcal{Y}^u$ and $\mathcal{V}$:

$$\begin{cases} \min_{\Psi} \max_{\Phi} \mathcal{L}^l(\Psi) + \mathcal{L}^u(\mathcal{Y}^u, \Psi, \mathcal{V}) + \beta \mathcal{L}^p(\Psi, \Phi), \\ s.t. \ \mathbf{V}_j = P(T(\mathbf{X}_j^u|\Psi)|\Phi), \forall j = 1, 2, \cdots, M, \end{cases} \qquad (7)$$

Similar to the optimization process of Eq.6, we first learn the parameters of the pace-generator by fixing $\Psi$:

$$\max_{\Phi} \mathcal{L}^p(\Psi, \Phi), \ \ s.t. \ \mathbf{V}_j = P(T(\mathbf{X}_j^u|\Psi)|\Phi), \qquad (8)$$

which encourages the pace-generator to predict the manually annotated labels from $\mathcal{D}^l$ as the realistic ones while predicting the predicted labels from both $\mathcal{D}^l$ and $\mathcal{D}^u$ as the fake ones. In the second stage, we learn the parameters of the task-predictor by fixing $\Phi$:

$$\begin{cases} \min_{\Psi} \mathcal{L}^l(\Psi) + \mathcal{L}^u(\mathcal{Y}^u, \Psi, \mathcal{V}) + \beta \mathcal{L}^p(\Psi, \Phi), \\ s.t. \ \mathbf{V}_j = P(T(\mathbf{X}_j^u|\Psi)|\Phi), \forall j = 1, 2, \cdots, M, \end{cases} \qquad (9)$$

which enables the task-predictor to learn informative patterns under the guidance of both the human annotated labels and the confident pseudo labels. The pace loss $\mathcal{L}^p(\Psi, \Phi)$ can also help explore the structure of the predicted labels to regularize the learning of task-predictor. For simplifying the optimization processes of Eq. 8 and Eq. 9, we loose their constraints by converting their constraints to the cross entropy-based loss terms in the learning objective functions.

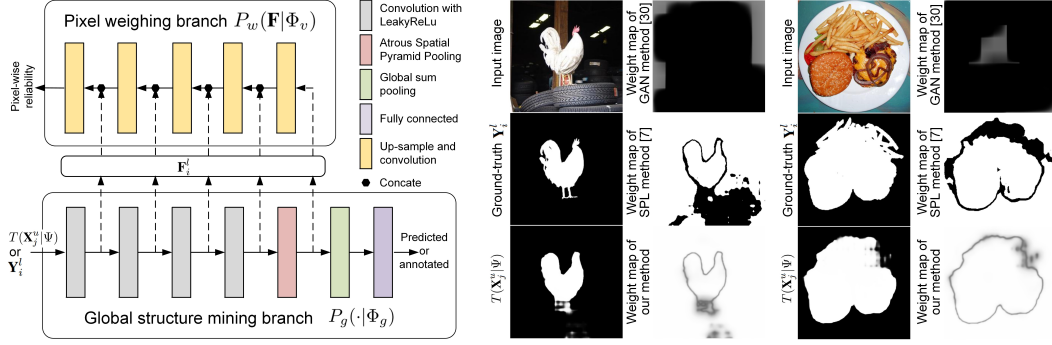

Figure 1: The proposed global structure-guided pixel weighting model and several visual comparison on the weight maps generated by our approach and the conventional GAN or SPL methods. The dashed lines indicate paths without back-propagation. Notice that the displayed weight maps are generated according to the predicted saliency mask $T(\mathbf{X}_j^u|\Psi)$ shown in the bottom-left corner of each set of examples. From the examples, we can observe that the proposed approach can effectively localize the unreliable object boundaries or background regions from the input saliency masks.

**Implementation for few-cost salient object detection.** We adopt the DeepLab-v2 [38] as the task-predictor $T(\cdot)$ by considering the trade-off between model effectiveness and computational cost. We further alleviate the memory cost by removing the multi-scale fusion module of DeepLab-v2.

When designing the pace-generator $P(\cdot)$, conventional methods, such as [35], might adopt the FCN architecture [39] with an up-sampling layer. They treat all regions of a generated saliency mask as unreliable regions and all regions of a ground-truth saliency mask as reliable regions and train the network parameters in an adversarial learning manner. However, as the generated saliency masks also contain reliable regions that are consistent with the ground-truth, such a learning manner would somehow mislead the learner and may obtain inaccurate weight map $\mathbf{V}_j$ (see Fig. 1). To this end, this paper designs a novel global structure-guided pixel weighting scheme which consists of a global structure mining (GSM) branch $P_g(\cdot)$ and a pixel weighting (PW) branch $P_w(\cdot)$. In the GSM branch, we first use four convolutional layers (with the kernel size of $4 \times 4$, channel number of $\{64, 128, 256, 512\}$, and stride of 2) to learn features. Then, we use a global sum pooling layer [40] followed by a fully connected layer to obtain a two-value vector as the prediction of whether the input mask is from model prediction or human annotation. Denote the network parameters in this branch as $\Phi_g$. We train $\Phi_g$ in an adversarial learning manner to learn global-structure patterns to infer the reliability of the input mask. To infer the finer pixel-wise reliability of the input mask, the PW branch takes the features extracted by the GSM branch as the inputs and is designed with a set of up-sampling blocks with skip connection to the previous convolutional layers (see Fig. 1). Denote the network parameters in this branch as $\Phi_v$. We train $\Phi_v$ in a supervised learning manner by using the ground-truth pixel-wise reliability $\mathbf{V}_i^*$ of the predicted saliency mask on the labeled training images:

$$\mathbf{V}_i^* = 1 - |T(\mathbf{X}_i^l|\Psi) - \mathbf{Y}_i^l|. \tag{10}$$

Then, Eq.5 becomes to:

$$
\begin{aligned}
\mathcal{L}^p &= \mathcal{L}^{p_g} + \mathcal{L}^{p_w}, \\
\mathcal{L}^{p_g} &= \sum\nolimits_{i \in \mathcal{D}^l} \log P_g(\mathbf{Y}_i^l|\Phi_g) + \sum\nolimits_{i \in \mathcal{D}^l} \log(1 - P_g(T(\mathbf{X}_i^l|\Psi)|\Phi_g)) \\
&\quad + \eta \sum\nolimits_{j \in \mathcal{D}^u} \log(1 - P_g(T(\mathbf{X}_j^u|\Psi)|\Phi_g)), \\
\mathcal{L}^{p_w} &= -\sum\nolimits_{i \in \mathcal{D}^l} \Gamma[(\mathbf{1} - \mathbf{V}_i^*) \log(\mathbf{1} - P_w(\mathbf{F}_i^l|\Phi_v)) + \mathbf{V}_i^* \log P_w(\mathbf{F}_i^l|\Phi_v)],
\end{aligned}
\tag{11}
$$

where $\mathbf{F}_i^l$ denotes the global structure features extracted by the GSM branch on $\mathbf{X}_i^l$. Finally, the entire learning objective function becomes to:

$$
\begin{cases}
\min\limits_{\Psi, \mathcal{V}, \mathcal{Y}^u, \Phi_v} \max\limits_{\Phi_g} \mathcal{L}^l(\Psi) + \mathcal{L}^u(\mathcal{Y}^u, \Psi, \mathcal{V}) + \beta \mathcal{L}^p(\Psi, \Phi_v, \Phi_g), \\
s.t. \ \mathbf{V}_j = P_w(\mathbf{F}_j^u|\Phi_v) = P(T(\mathbf{X}_j^u|\Psi)|\Phi_v, \Phi_g), \forall j = 1, 2, \cdots, M.
\end{cases}
\tag{12}
$$

The optimization process of Eq. 12 still follows the pseudo algorithm shown in Alg. 2. The only difference is that when updating $\Theta = \{\Psi, \Phi_g, \Phi_v\}$ with fixed $\mathcal{Y}^u$ and $\mathcal{V}$, we optimize the following function for instead:

$$\min_{\Psi, \Phi_v} \max_{\Phi_g} \mathcal{L}^l(\Psi) + \mathcal{L}^u(\mathcal{Y}^u, \Psi, \mathcal{V}) + \beta \mathcal{L}^p(\Psi, \Phi_g, \Phi_v). \qquad (13)$$

## 4    Experiments

We use four widely-used benchmark datasets to implement the experiments, which include PASCAL-S [41], DUT-O [42], SOD [43], and DUTS [28]. Following the previous works [44, 20, 45], we use the training split of the DUT-S dataset for training and test the trained models on the other datasets. Notice that different from the previous works that require the pixel-wise manual annotation on every training images, the approach presented in this work only needs the pixel-wise manual annotation for 1k training images, which is about one-tenth of the whole training images. We use the F-measure and mean absolute error (MAE) to evaluate the experimental results.

We implement the proposed algorithm on the PyTorch framework using a NVIDIA GTX 1080Ti GPU. When training the saliency network, we use the Stochastic Gradient Descent (SGD) optimization method, where the momentum is set to 0.9, and the weight decay is set to $5 \times 10^{-4}$. The initial learning rates of the task-predictor and the pace-generator are $2.5 \times 10^{-4}$ and $10^{-4}$, respectively, which are decreased with polynomial decay parameterized by 0.9. For training the pace network, we adopt the Adam optimizer [46] with the learning rate $10^{-4}$. The same polynomial decay as the saliency network is also used. We set $\beta = 0.01$ and $\eta = 0.7$ according to a heuristic grid search process. our method uses in total 24.5K iterations and the loss and performance curves are shown in Fig. 2.

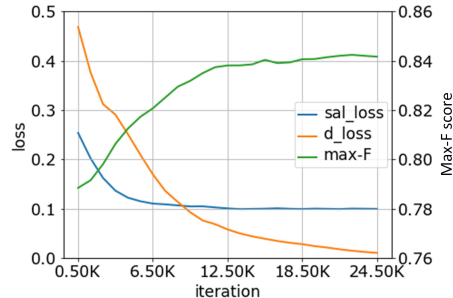

Figure 2: Loss and performance curves on the training split of the DUT-S dataset.

### 4.1    Comparison to the State-of-the-Art Salient Object Detection Methods

In this section, we compare the proposed approach with 12 state-of-the-art salient object detection methods, which contain both the fully supervised state-of-the-art methods [47, 48, 26, 49, 45, 25, 50, 51] and the unsupervised or semi-/weakly supervised ones [52–54, 28, 34, 55, 56][1]. Notice that all the compared salient object detection models are trained on the same training set but with different amounts or forms of manual annotation. For the fully supervised models, pixel-wise manual annotation of all the training images (about 10k training images) is required. For the unsupervised and weakly supervised models, none pixel-wise manual annotation is required but a certain scope of the image-level annotation is needed. In contrast, the proposed few-cost model uses the pixel-wise manual annotation on only 1k training images. Thus, our approach has much less annotation cost than the conventional fully supervised methods but slightly larger annotation cost than the unsupervised or weakly supervised methods.

The comparison results between our approach and these state-of-the-art salient object detection methods are reported in Table 1 and Fig, 3. From the experimental results, we can observe that our approach outperforms all the state-of-the-art unsupervised or weakly supervised salient object methods with noticeable performance gains. When compared with the existing fully supervised salient object detection methods, our approach can effectively approach the most state-of-the-art method and obtains even better results than some of them. This demonstrates the effectiveness of the proposed approach and implies the rationality of the investigated few-cost salient object detection task in addressing the annotation-hunger issue of the existing salient object detection methods. Some qualitative comparisons of the annotation results are also shown in Fig. 3.

Table 1: Comparison of the proposed approach with the state-of-the-art salient object detection methods as well as our baseline models on the PASCAL-S, DUT-O, SOD, and DUT-TE datasets.

| Methods | | DUTS-TE | | PASCAL-S | | DUT-O | | SOD | |
|---|---|---|---|---|---|---|---|---|---|
| | | $F_{max}$ | MAE | $F_{max}$ | MAE | $F_{max}$ | MAE | $F_{max}$ | MAE |
| Fully supervised SOD | F3NET | **0.897** | **0.035** | 0.878 | **0.061** | 0.839 | **0.053** | – | – |
| | EGNet[47] | 0.893 | 0.039 | 0.869 | 0.074 | **0.842** | **0.053** | **0.889** | **0.099** |
| | AFNet[48] | 0.867 | 0.045 | 0.866 | 0.070 | 0.820 | 0.057 | – | – |
| | PoolNet[26] | 0.894 | 0.036 | **0.884** | 0.065 | 0.830 | 0.054 | 0.879 | 0.106 |
| | BASNet[49] | 0.860 | 0.047 | 0.858 | 0.076 | 0.811 | 0.056 | 0.851 | 0.114 |
| | BRN[45] | 0.828 | 0.049 | 0.849 | 0.072 | 0.774 | 0.062 | 0.846 | 0.105 |
| | MLMSNet[25] | 0.854 | 0.048 | 0.858 | 0.074 | 0.793 | 0.064 | 0.862 | 0.108 |
| | PAGE-Net[50] | 0.838 | 0.051 | 0.850 | 0.076 | 0.791 | 0.062 | 0.842 | 0.111 |
| | PAGRN[51] | 0.854 | 0.055 | 0.849 | 0.089 | 0.771 | 0.071 | 0.838 | 0.147 |
| Un-/semi/ weakly supervised SOD | MWS[52] | 0.768 | 0.091 | 0.786 | 0.134 | 0.722 | 0.108 | 0.801 | 0.170 |
| | ASMO[53] | – | – | 0.758 | 0.154 | 0.732 | 0.100 | 0.758 | 0.187 |
| | C2S-NET[54] | 0.807 | 0.062 | 0.842 | 0.082 | 0.758 | 0.072 | – | – |
| | WSS[28] | 0.740 | 0.099 | 0.773 | 0.140 | 0.695 | 0.110 | 0.778 | 0.171 |
| | SGAN[34] | 0.610 | 0.135 | 0.699 | 0.164 | 0.610 | 0.131 | – | – |
| | DeepUSPS[55] | – | – | – | – | 0.736 | 0.063 | – | – |
| | SODSA[56] | 0.789 | 0.062 | 0.811 | 0.092 | 0.753 | 0.068 | 0.806 | 0.131 |
| FC-SOD | Ours | 0.846 | 0.045 | 0.848 | 0.067 | 0.767 | 0.067 | 0.846 | 0.122 |

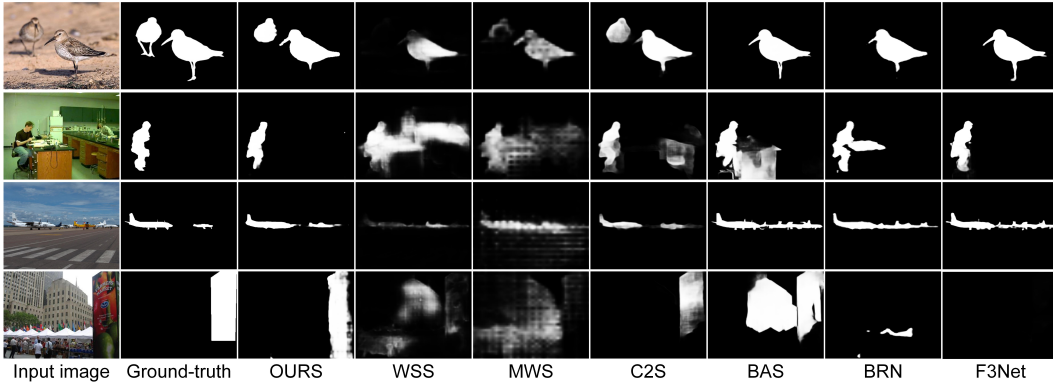

Input image  Ground-truth  OURS  WSS  MWS  C2S  BAS  BRN  F3Net

Figure 3: Some examples of the saliency detection results obtained by our approach and other state-of-the-art methods.

## 4.2 Comparison to the SPL Methods and the Ablation Study Models

In this section, we compare our approach with five self-paced learning schemes [7, 9, 36, 10, 57]. To implement these SPL methods, we replace the $\mathcal{L}^p$ term in our learning object function with their proposed self-paced regularizers and adopt their optimization procedures to train the task-predictor. All other settings are kept the same as the proposed approach. In table 2, we show the comparison results on the DUTS-TE and DUT-O datasets. From the comparison results, we can observe that with the heuristically-designed self-paced regularizers, the conventional SPL methods cannot work well on the investigated task. This indicates the human knowledge embedded in the existing self-paced regularizers is insufficient or inaccurate for few-cost salient object detection.

Besides, we also conduct ablation studies by comparing our approach with five baseline models. The first baseline only uses the $\mathcal{L}^l$ term of Eq. 2, which trains the saliency network by only using the manually annotated training images. The second baseline introduces both the labeled data and unlabeled data in training and considers the pseudo-labels on the unlabeled training images with equal reliability. Based on this baseline, the "Grab-cut" baseline further uses a naive strategy to refine the generated pseudo-labels by using GrabCut [58]. The "Pixel GAN" baseline adopts the conventional pixel GAN [35, 59] to formulate the $\mathcal{L}^p$ term to facilitate the learning process. In this

Table 2: Comparisons to the SPL models.

| Methods | DUTS-TE | | DUT-O | |
|---|---|---|---|---|
| | $F_{max}$ | MAE | $F_{max}$ | MAE |
| $\ell_1$-based [7] | 0.817 | 0.056 | 0.732 | 0.081 |
| LiS-based [9] | 0.819 | 0.055 | 0.732 | 0.081 |
| $\ell_{0.5,1}$-based [36] | 0.818 | 0.056 | 0.731 | 0.080 |
| $\ell_{2,1}$-based [10] | 0.820 | 0.055 | 0.733 | 0.079 |
| Fraction-based [57] | 0.818 | 0.055 | 0.729 | 0.080 |
| **Ours** | **0.846** | **0.045** | **0.767** | **0.067** |

Table 3: Ablation studies.

| | DUTS-TE | | DUT-O | |
|---|---|---|---|---|
| | $F_{max}$ | MAE | $F_{max}$ | MAE |
| Only $\mathcal{L}^l$ | 0.817 | 0.059 | 0.719 | 0.085 |
| w/o $\mathcal{L}^p$ | 0.822 | 0.054 | 0.736 | 0.078 |
| GrabCut | 0.730 | 0.120 | 0.616 | 0.129 |
| Pixel GAN | 0.842 | 0.046 | 0.754 | 0.071 |
| w/o V* | 0.840 | 0.046 | 0.752 | 0.068 |
| **Ours** | **0.846** | **0.045** | **0.767** | **0.067** |

case, our learning objective function degenerates to Eq. 2. Like in Pixel GAN, the "w/o V*" baseline learns our PW branch by constraining all pixels in the predicted saliency masks as the unreliable ones while all pixels in the ground-truth saliency masks as the reliable ones. From the experimental results reported in Table 3, we can observe that simply using the naive GrabCut strategy cannot improve the quality of the generated pseudo-labels. Instead, it would introduce additional noise to the pseudo-labels. Compared to the conventional pixel GAN-based formulation, our approach can better infer the reliability weights for the generated pseudo-labels. It is also interesting to see that the convention SPL methods cannot improve the learning performance of the "w/o $\mathcal{L}^p$" baseline. To our best knowledge, this might due to the conventional SPL methods are limited in exploring the structure of the saliency masks and the weight maps obtained by them would hurt the structure of the salient object regions (see examples in Fig. 1).

We have also carried out experiments under different ratios of labeled data with the goal to validate our proposed method's robustness. The experimental results are reported in Table 4. As can be seen, our approach can learn with different ratios of labeled data. Notably, when only using 1 percent training labels, our approach can still achieve 95.48% performance (in terms of maxF) of the model trained on full training labels.

Table 4: Experiments under different ratios of labeled data.

| | 1% | 5% | 10% | 30% | Full |
|---|---|---|---|---|---|
| max F | 0.824 | 0.840 | 0.846 | 0.846 | 0.863 |
| MAE | 0.054 | 0.049 | 0.045 | 0.044 | 0.044 |

## 5 Conclusion

This paper studied a problem called few-cost salient object detection. Unlike the conventional salient object detection methods that require large amounts of human annotation, FC-SOD uses only the manual annotation on a few training images together with the annotation-free images. Specifically, we propose an adversarial-paced learning (APL)-based framework to facilitate the few-cost learning scenario. Essentially, APL is derived from the self-paced learning (SPL) regime but it infers the robust learning pace through the data-driven adversarial learning mechanism. For implementing APL for FC-SOD, we further design a global structure-guided pixel weighting scheme to infer the reliability weights for image regions. Comprehensive experiments on widely-used benchmark datasets have demonstrated the effectiveness of the proposed approach. Notably, by using the annotation of only 1k training images, our approach can outperform the existing weakly supervised SOD approaches and perform comparably to the fully supervised SOD models.

## Broader Impact

To our best knowledge, this research would provide intelligent visual perception to assistive robotics, which might offer supports in allowing people to live healthier and independent lives for longer. It is believed that advances in automatic saliency detection are net positive for society, despite the potential for misuse. The consequences of the failure of the system would lead to a false understanding of the image content. The task and method do not leverage biases in the data.

## Footnotes

[1]We also intend to compare to [6]. However, we are not able to acquire their model or detection results on the datasets used in our comparison.

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
