[Reviews · NeurIPS 2020]

Review 1

Summary and Contributions: In this paper, the authors addressed a new problem setting called the few-cost salient object detection (FC-SOD). In FC-SOD, the training data consist of a few annotated images and a lot of unlabeled images, and in this way, the annotation cost could be largely reduced. Besides, to address the proposed problem, a novel adversarial paced learning framework is established by integrating the self-paced learning into the generative adversarial learning. In this framework, the robust learning pace could be reliably inferred with an additional model called the pace-generator. The iterative optimization scheme is adopted to optimize the task-predictor and the pace-generator. The experiments on widely-used benchmark datasets are evaluated, and the proposed method outperforms the existing un-/semi-/weakly learning saliency detection. The abundant ablation studies are also included to validate the performance gain of the proposed method.

Strengths: 1. The addressed task, few-cost saliency detection, is interesting and worth being explored. The problem setting is also new. 2. The proposed approach, which combines self-paced learning, is novel and technically sound. 3. The paper is clearly presented and well-organized. 4. The proposed method outperforms the current state-of-the-art method of unsupervised or weakly supervised saliency detection. Many ablation studies are conducted to validate the contribution of the proposed components.

Weaknesses: 1. Some related works missing There are some recent related works, such as [Ref. 1~Ref.6]. In [Ref. 1, 2], the authors integrated the self-paced learning into the object co-saliency detection related to the addressed task of this paper. These two are close to the proposed work, and it is better to provide the discussion. In [Ref. 3, and 4], two latest works of weakly or unsupervised saliency detection, it is better to cite and compare these two papers. In [Ref 5, and Ref. 6], the authors also combine self-paced learning and adversarial learning, and I think these two works are mostly related to the proposed method, and I would like to see the difference between the proposed method and [Ref 5 and 6] Besides, in [3, 23, 27], semi-supervised learning for saliency detection is addressed, but in this paper, there is no detailed discussion between semi-supervised learning [23, 27] and the proposed few-cost setting. The primary difference should be provided. [Ref. 1] Tsai et al., "Deep Co-saliency Detection via Stacked Autoencoder-enabled Fusion and Self-trained CNNs," TMM'19 [Ref. 2] Hsu et al., "Unsupervised CNN-based Co-Saliency Detection with Graphical Optimization," ECCV'18 [Ref. 3] Zhang et al., "Weakly-Supervised Salient Object Detection via Scribble Annotations," CVPR'20 [Ref. 4] Nguyen et al., "DeepUSPS: Deep Robust Unsupervised Saliency Prediction With Self-Supervision," NIPS'19 [Ref. 5] Zhang et al., "Self-Paced Collaborative and Adversarial Network for Unsupervised Domain Adaptation," PAMI'19 [Ref. 6] Dizaji et al., "Balanced Self-Paced Learning for Generative Adversarial Clustering Network" CVPR'20 2. Similar to the semi-supervised segmentation Although the problem setting is new, the semi-supervised segmentation is similar to the proposed problem setting. The training data also contains a few manual annotated images and a lot of annotated images in the semi-supervised segmentation, such as [24, 25, 28]. The difference between the salient object and the semantic segmentation is the number of the classes. The salient object is a binary problem, and the semantic segmentation is a multi-class problem. Therefore, these semi-supervised segmentation algorithms could be easily applied to few-cost salient detection. Therefore, if we don't consider the number of classes, what is the difference between the proposed few-cost salient object detection and the semi-supervised segmentation? 3. Ablation Study 2-1. How many iterations are used in the proposed method? Besides, because the iteration method is used to optimize the proposed loss, the convergence should be ensured. Therefore, the losses and the performance under the different iterations should be given and compared. 2-2. Because the address in [3] is similar to this work, the comparison should be added in Tabel 3. 2-3. The experiments under the different ratios of annotated ground-truth should be conducted to validate the proposed method's robustness. The study is usually done in the previous semi-supervised segmentation.

Correctness: Yes.

Clarity: Yes.

Relation to Prior Work: No. Please refer to Weaknesses

Reproducibility: Yes

Additional Feedback:


Review 2

Summary and Contributions: This paper aims to reduce the labeling cost for data driven single image salient object detection. It argues that it can use only a few labeled data for training. It also argues that the proposed approach is not semi-supervised learning but should be named as few-cost learning because semi-supervised learning in salient object detection is to partially label the in-image region rather than the image samples.

Strengths: Overall the paper is complete and easy to follow.

Weaknesses: The contribution is very limited, there is not particular insights on the proposed methods on salient object detection. It only applied well explored gradually learning approach to salient object detection. The approach can be applied to any machine learning tasks. Throughout the first part of Section 3, I do not see any insight while the proposed adaptive method has anything particular to do with salient object detection. Further more, I don't find section 3 even start to define the salient object detection problem before the authors pave through the equations. Quote "Given a small-scale collection Dl, which consists of the manually labeled training images ..., and a larger-scale collection Du that contains the unlabeled training images ...". This is a quite standard statement for any semi supervised learning emthod. The argument and definition of few-cost learning against semi-supervised learning and weakly supervised learning is not correct. The experiment is insufficient, while the key argument of this paper is the Few-Cost, I could not find any study about the cost. It seems all the experiments are run on a default number of labeled images. I could not find the default number either. Lastly, all the references are from computer vision conferences, it raises the concern that this paper should be submitted to a vision conference, rather than NeurIPS.

Correctness: Looks reasonably correct.

Clarity: Yrd

Relation to Prior Work: No. The argument and definition of few-cost learning against semi-supervised learning and weakly supervised learning is not correct.

Reproducibility: Yes

Additional Feedback:


Review 3

Summary and Contributions: This paper addresses the label cost issue in the research field of salient object detection. The goal is to learn the desired deep salient object detector by only using a small number of labeled training images. To achieve is goal, the authors build an adversarial-paced learning-based framework. The basic idea is to learn to label the unlabeled training images and select reliable image regions for guiding the subsequent learning process. Four benchmark datasets are used in the experiments to provide a comprehensive evaluation of the proposed method.

Strengths: +Salient object detection is a popular topic in recent years. Many existing works are motivated to achieve superior performance. On the contrary, this work studies a under-explored yet important problem in this field, i.e., the label cost issue. To some extent, this work looks similar to the semi-supervised method. But as the authors have pointed out that the existing semi-supervised SOD methods consider different problems with this work. Thus, I think this paper presents an interesting and valuable research direction to the field of SOD. +The APL is another interesting point of this work. It looks like a kind of combination of SPL and GAL. But what is interesting to me is that it solves a critical problem in SPL, which is the hand-crafted design of the self-paced regularization term. In fact, to my knowledge, traditional self-paced regularization terms need to be designed based on the human observation/understanding of the data. While the APL presented in this paper can do this in a data-driven manner.

Weaknesses: -Table 1 shows that the proposed FC-SOD achieves better performance than the weakly/semi-/unsupervised methods. I am curiousness about whether it is a common situation or it is because of the newly designed learning framework. In other words, the authors are encouraged to discuss what are the upper bounds of the weakly, semi- and few cost SOD methods? With the same label cost, which one should have the higher upper bound? -In the ablation study, the authors should report the upper bound performance of the proposed method, i.e., train the used deep salient object detector under a fully supervised manner. -The authors should discuss whether the existing semi-supervised semantic segmentation approaches can be directly adopted to address the problem investigated in this work. On the other hand, the authors should also discuss the potential of using the proposed APL framework to address the semi-supervised semantic segmentation problem. -Could the authors provide some hints to explain why the proposed approach can sometimes outperform the fully supervised SOD methods?

Correctness: Yes. The claims meet my background well and I can follow them easily.

Clarity: yes. The methodology descriptions, figures, and tables are all well organized and easy to follow.

Relation to Prior Work: Yes. The authors have provided a clear comparison between this work and the previous SOD works and semi-supervised GAL works.

Reproducibility: Yes

Additional Feedback:

[Author Response · NeurIPS 2020]

Thanks all reviewers for their thoughtful comments. Below, we conduct a point-point response to each comment.

**R2.1 Missing related works:** We will discuss them in the revised manuscript. Briefly speaking, [Tsai et al. TMM19] and [Hsu et al. ECCV18] apply the SPL processes to refine the saliency maps obtained in co-saliency detection. [Zhang et al. CVPR20] and [Nguyen et al. NIPS19] are two latest works on weakly and unsupervised saliency detection and we will report their performance in the revised manuscript. [Zhang et al, PAMI20] and [Dizaji et al. CVPR20] integrate SPL and AL for **domain adaption** and **clustering**, respectively. They adopt **heuristically designed** self-paced regularizers to guide the AL processes. In contrast, the self-paced regularizer in our model is **implicitly learned in a data-driven manner** rather than being heuristically designed and it guides an FCL process rather than an AL process.

**R2.2 Discussion with [23] and [27]:** As we mentioned in the paper, these works share a similar spirit with our work. As for the difference, Yan et al. [23] define their problem on sparsely annotated video frames. The learning process is based on the dependencies among adjacent video frames. In contrast, our task does not have such dependency to explore. [27] presents an IoT-oriented saliency learning framework. The intention is to leverage both labeled and unlabeled data from different problem domains. It requires to generate image data for different problem domains. In contrast, our task works under a single and common problem domain, thus needing no such data generation process.

**R2.3 Relation between semi-supervised segmentation (SSS) and FCSOD:** Besides the superficial difference in the number of classes, the challenges met by them are also different. Specifically, as the salient class that needs to be segmented in FCSOD would usually cover a number of different semantics rather than forming by a specific semantic, FCSOD is encountered with heavier **intra-class variance** than SSS. This would bring the challenging learning ambiguity issue in FCSOD. Besides, the current SSS methods usually leverage the semantic vector as an informative attribute to guide the GAN-based semi-supervised learning process, which, however, is **absent** from the FCSOD task. Consequently, such SSS algorithms could not be easily applied to FCSOD.

**R2.4 Ablation study:** We will add more ablation studies into the revised paper. Specifically, our method uses in total 24.5K iterations and the loss and performance curves are shown in Fig. 1. For [3], we will report its results in the revised paper. We have also carried out experiments when varying ratios of labeled data. As reported in Table 1, our approach can learn with different ratios of labeled data robustly.

Table 1. Experiments under different ratios of labeled data.

|      | 1%   | 5%   | 10%  | 30%  | Full |
|------|------|------|------|------|------|
| maxF | .824 | .840 | .846 | .846 | .863 |
| MAE  | .054 | .049 | .045 | .044 | .044 |
| S    | .793 | .814 | .822 | .831 | .854 |

Fig. 1. Loss and performance curves.

**R3.1 Contribution and novelty:** As recognized by recent works in the SOD community [20-23], one important issue is that the annotation of salient regions is very tedious and training samples with accurate annotations remain scarce and expensive. To solve this problem, we study an under-explored yet meaningful learning scenario and propose a novel learning framework. Thus, we do believe that we contribute to the saliency detection community, as evidenced by the statements of the other two reviewers, e.g., "valuable research direction to the field of SOD", "worth being explored", etc. Besides, we kindly argue that we build a novel APL framework rather than "applying a well-explored gradually learning approach". Featured by the capacity to infer learning pace under a data-driven adversarial learning manner, the proposed APL is beyond the exploration of any existing work.

**R3.2 Argument of FCL against SSL and WSL:** In the context of SOD, we usually treat each image as a sample. As the learning problem that uses a few samples to train a target model is named as few-shot learning, we name our problem as FCL as it requires annotation cost only on **a few** samples. Different from FCSOD, SSSOD provides **incomplete annotation for every sample**, while WSSOD provides **weak (image-level) annotation for every sample**. Under this circumstance, we think our argument of FCL against SSL and WSL is correct.

**R3.3 Cost study:** We kindly remind the reviewer that we have actually provided the number of labeled images in L16&245, which is 1K. Compared to the traditional SOD methods that train on 10K labeled images, our approach only requires 10% annotation cost, thus called few-cost SOD. In addition, we have conducted more experiments to study the performance of our approach under various annotation costs as reported in Table 1.

**R3.4 More fit to vision conferences:** Our core innovation is the novel adversarial-paced learning scheme, which we believe fits NeurIPS well. Besides, we kindly remind the reviewer that many vision task-related works have been published in NeurIPS, such as the famous Faster R-CNN [Ren et al. NeurIPS15] and Object bank [Li et al. NeurIPS10], and the recent SOD methods [Nguyen et al. NeurIPS19] and [Zhang et al. NeurIPS19].

**R4.1 Upper bound discussion:** This is an interesting point. To our best knowledge, the upper bound of FCSOD should be higher than unsupervised SOD as unsupervised SOD does not use any human annotation. With the same label cost, the upper bound of FCSOD and WSSOD should be close. However, as FCSOD leverages a small number of strong annotations while WSSOD leverages a lot of weak annotations, FCSOD should theoretically work better when dealing with data with small domain shifts.

**R4.2 Fully supervised performance of the used saliency model:** Please see the last column of Table 1.

**R4.3 Whether SSS methods can be directly adopted to FCSOD:** As we discussed in **R2.3**, they cannot.

[Meta-Review · NeurIPS 2020]

This paper received reviews from 3 expert reviewers. The reviewers appreciated the interesting task (few cost saliency detection) and the use of self-paced learning combined with generative adversarial learning. After considering the authors' response, the reviewers refined their positions on the paper. R2's comments regarding semi-supervised learning remain valid. The authors would be encouraged to refine the presentation of this and use of terms. While I understand that the specific task may have its own nuances, it very much seems to be an instance of semi-supervised learning as R2 notes -- it includes labeled and unlabeled data and one learns from these. The descriptions that point to the contrary in the text seem unnecessary. Overall, given the interesting algorithmic contribution and good empirical results, this paper would be appropriate for presentation at NeurIPS.